

# The surface albedo of the Greenland Ice Sheet between 1982 and 2015, and its relationship to the ice sheet's surface mass balance and ice discharge

Aku Riihelä[1], Michalea D. King[2], Kati Anttila[1]

[1]Finnish Meteorological Institute, Helsinki, FI-00560, Finland
[2]Byrd Polar and Climate Research Center, Columbus, USA.

*Correspondence to*: Aku Riihelä (aku.riihela@fmi.fi)

**Abstract.** The Greenland Ice Sheet is losing mass at a significant rate, primarily driven by increasing surface melt-induced
runoff. Because the ice sheet's surface melt is closely connected to changes in the surface albedo, studying multidecadal
changes in the ice sheet's albedo offers insight into surface melt and associated changes in its surface mass balance. Here, we
first analyse the CLARA-A2 SAL satellite-based surface albedo dataset, covering 1982-2015, to obtain decadal albedo trends
for each summer month. We also examine the rates of albedo change during the early summer, supported with atmospheric
reanalysis data from MERRA-2, to discern changes in the intensity of early summer melt, and their likely drivers. We find that
rates of albedo decrease during summer melt have accelerated during the 2000s relative to early 1980s, and that the surface
albedos now often decrease to values typical of bare ice at elevations $50 - 100$ m higher on the ice sheet. The southern margins
exhibit the opposite behaviour, though, and we suggest this is due to increasing snowfall over the area.

We then correct the mass balance estimates observed by the GRACE satellite mission with state-of-the-art ice discharge
estimates to obtain observation-based estimates for the surface mass balance. The CLARA albedo changes are regressed with
this data to obtain a proxy surface mass balance timeseries for the summer periods 1982-2015. This proxy timeseries is
compared with latest regional climate model estimates from the MAR model. We show that the proxy timeseries agrees with
MAR through the analyzed period within the associated uncertainties of the data and methods, demonstrating and confirming
that surface runoff has dominated the rapid mass loss period between 1990s and 2010s.

Finally, we extend the analysis to GrIS basin scale to examine discharge-albedo relationships in order to ascertain if the surface
melt contributes to discharge acceleration via basal lubrication. While there is little evidence of surface melt-induced ice flow
acceleration at annual timescales, we find time lags between seasonal maximum runoff production and seasonal maximum
discharge rate to be in agreement with recent modelling results.

## 1.  Introduction

The surface albedo of the Greenland Ice Sheet's ablation area has been shown to play a crucial role in determining melt
variability through the snow/ice albedo feedback effect (Box et al., 2012). The study of these albedo changes over the last



decade and a half from modern satellite sensors such as MODIS has drawn considerable research attention (Tedesco et al., 2011; Stroeve et al., 2013; Alexander et al, 2014; Tedstone et al., 2017). The observed albedo decreases are closely tied to the increasing mass loss of the ice sheet through the enhancement of its surface melt (Enderlin et al., 2014; van den Broeke et al., 2016). The ice sheet also loses mass through the ice calving at outlet glaciers, or discharge, the magnitude of which has recently been shown to demonstrate significant spatial and seasonal variability (King et al., 2018). The mass loss of the Greenland Ice Sheet (henceforth GrIS) has been shown to currently be among the leading contributors to sea level rise (Box et al., 2017).

Legacy satellite sensors such as AHVRR offers us the possibility to study the GrIS albedo changes of over a period of more than three decades, doubling the temporal coverage of MODIS. The drawback is lower spectral precision and coverage. Yet, considerable efforts have recently been paid to the intercalibration of the AVHRR sensor family (Heidinger et al., 2010), greatly improving the sensors' capabilities to detect long-term trends in surface albedo. Here, we use the sensor-intercalibrated 34-year (1982-2015) surface **a**lbedo data record from the CLARA-A2 dataset family (Karlsson et al., 2017) to study the albedo changes of the ice sheet at decadal scales.

Our analyses focus on the summer months between May and August (MJJA), comprising four topics. First, we calculate and present the trends in surface albedo (defined here as the directional-hemispherical reflectance of the surface, i.e. black-sky albedo or 'inherent' albedo) over the GrIS for each respective month. In addition to the trends for the full 1982-2015 period, we present temporally subsetted trends consistent with the MODIS observation era (2000-2015) and the pre-MODIS era (1982-1999), in part to compare our results with the trends identified from the latest editions of MODIS data (Casey et al., 2017) and also to offer a rare, if somewhat more uncertain, look into the ice sheet's surface changes in the 80s and 90s. Treating each month individually additionally allows us to delineate between albedo changes in the early and late melting season.

Second, we examine changes in melt intensity and speed through changes in the rate of albedo decrease and in the spatial distribution of albedo values indicative of bare ice and wet snow as the summer melt progresses. At this stage we also analyse atmospheric reanalysis data to discern if the observed changes may be explained by precipitation or air temperature anomalies.

Third, we show that, when calibrated against discharge-corrected state-of-the-art gravimetric observations of the ice sheet, the CLARA surface albedo estimates can serve as a useful stationary proxy for the summer surface mass balance (SMB) changes during its 34-yr coverage. The SMB estimates achieve good agreement with the MAR regional climate model (Fettweis et al., 2017) for the study period, both highlighting and confirming the dominant role of surface melt in the recent acceleration of the GrIS summer mass loss.





Finally, we downscale the analysis to six major drainage basins of the ice sheet. Having demonstrated that surface albedo changes are an acceptable proxy for surface runoff production, we examine the albedo-discharge relationships to ascertain from observational basis if inferred surface melt anomalies correlate to seasonal maximum ice export.

## 2. Data and Methods

### 2.1. CLARA-A2 SAL surface albedo dataset

The satellite-based surface albedo data used here is the Satellite Application Facility on Climate Monitoring (CM SAF) cLoud, Albedo and surface RAdiation data record CLARA-A2 SAL, funded by the European Organization for the Exploitation of Meteorological Satellites EUMETSAT. It covers the years 1982-2015 and is based on intercalibrated AVHRR data. The retrieved albedo is defined as directional-hemispherical albedo for the wavelength range 0.25 -2.5 µm (for snow and ice), and the observations are provided in a 0.25° latitude-longitude or a 25 km EASE-2 grid for the Polar Regions. The algorithm accounts for topography corrections over mountainous regions as well as dynamic aerosol depth (Jääskeläinen et al., 2017). However, over the Arctic where dynamic aerosol loading is difficult to estimate and typically exhibits a limited variability, a constant value of 0.1 at 550 nm is used.

The albedo of snow- and ice-covered areas is derived by averaging the broadband bidirectional reflectance values of the AVHRR overpasses into 5-day or monthly means, relying on wide AVHRR swaths and temporal aggregation to provide dense sampling of the viewing hemisphere, which forms the albedo estimate (Riihelä et al., 2013). The sampling rate has been further improved by the expansion of the AVHRR constellation from the mid-1990s onward. The CLARA dataset uses data from NOAA-7 to NOAA-19, as well as METOP A and B (Karlsson et al., 2017). A critical feature of the CLARA dataset family is that it is based on an intercalibrated Fundamental Climate Data Record (FCDR) of observed AVHRR radiances, where the calibration differences between satellites have been removed by use of a variety of techniques (Heidinger et al., 2010).

The data record has been validated against in situ data (Anttila et al., 2016) and compared to the MODIS black sky albedo product MCD43 (Schaaf et al., 2002). Comparisons between CLARA-A2 SAL and in situ measurements on GrIS have yielded RMSE between 0.04 and 0.07 over the more stable accumulation zone with no obvious systematic over- or underestimation tendencies, and a RMSE of ~0.11 over a site (JAR2) in the ablation zone. It should be noted that it has recently been shown that point-to-pixel evaluation of surface albedo retrievals over the heterogeneous surfaces of GrIS ablation zone, particularly in the southwest, cannot be expected to yield true estimates of retrieval quality when the spatial resolution of the estimate is in the kilometre range (Moustafa et al., 2017), as is certainly the case for CLARA-A2 SAL.

A critical attribute for the surface albedo timeseries from the viewpoint of trend determination is its stability. Here, we demonstrate the stability of CLARA-A2 SAL by examining the monthly mean albedo over a small region in the central part



of GrIS between 73 and 75 degrees North latitude, and 38 and 42 degrees West longitude. The innermost parts of GrIS have a naturally stable surface albedo, and therefore variations in the albedo estimates may be considered to primarily result from algorithm uncertainty. The mean surface albedo for this area is shown as a timeseries in Figure 1 for the months between May and August (MJJA). As can be seen, the variability about the 34-yr interannual mean largely falls within the reported 5%

(relative) accuracy, although individual grid cell-scale albedo retrievals may display larger uncertainty as a function of cloud masking efficiency, surface sampling rate, and uncertainty in the atmospheric correction of the satellite imagery. Still, we observe no obvious trends in the albedo of this region. The widespread surface melt of 2012 was relatively short-lived over this area (Bennartz et al., 2013), and thus did not appreciably impact the albedo.

A notable deviation from the interannual mean occurs over the years 1992 and 1993. The cause of this is known. The eruption of Mt. Pinatubo on June 15, 1991, injected vast quantities of aerosols and dust into the atmosphere. This altered the atmospheric composition on a global scale for the next 2-3 years, and is manifested here as a ~0.02 overestimation of the GrIS albedo during these years. Therefore, we henceforth exclude these years from the analysis but otherwise conclude that the timeseries appears stable enough for trend analysis. The observed variability envelope shall form the first, but not only, boundary

condition for evaluating the significance for any obtained trends, as explained in detail in section 2.7.2.

To exclude the non-glaciated surfaces of Greenland from the analysis, the PROMICE ice mask (Citterio and Ahlstrøm, 2013) was resampled into the CLARA grid and all grid cells with a fractional ice cover of less than 50% were discarded. We also attached surface elevation data into the CLARA dataset from CLARA grid-resampled DEM by Helm et al. (2014).

### 2.2. MAR regional climate model

The regional climate model data comes from the Modèle Atmosphérique Régional (MAR) model (version 3.5.2) (Fettweis et al., 2017). Here, we use the model run data forced with the ERA-Interim atmospheric reanalysis for the period 1979-2014. To obtain the MJJA surface mass balance from MAR, we first extract the monthly surface mass balance fields, correct for the variation in grid cell areas in the model domain, mask out all grid cells with fractional ice cover less than 50% for consistency

with the CLARA data, and sum the MJJA output.

### 2.3. GrIS ice discharge estimates

Here we use continuous estimates of ice discharge from all large Greenland outlet glaciers over the 2000-2015 period, taken from King et al. (2018). These glacier-scale time series are derived by incorporating changes in glacier velocity, from both radar and optical imagery, and changes in glacier thickness calculated from time-varying surface elevation data. With high

temporal and spatial resolution, these records allow for GrIS-wide and basin-scale sub-monthly changes in ice discharge, hereafter referred to as D, to be compared with albedo-based meltwater proxies.





### 2.4. MERRA-2 atmospheric reanalysis and the Greenland Blocking Index (GBI)

The Modern-Era Retrospective analysis for Research and Applications, Version 2 (MERRA-2) is a global atmospheric reanalysis of the modern satellite era, covering the period from 1980 to present day (Gelaro et al., 2017). Operating at 72 atmospheric levels with a spatial resolution of 0.5 by 0.625 degrees, MERRA-2 ingests a wide variety of satellite observations

and features e.g. observation-based corrections of the model precipitation field and a sophisticated treatment for atmospheric aerosols.

The representation of glaciated surfaces has also received substantial attention in the development of MERRA-2 (Cullather et al., 2014). It was also recently shown that the surface air temperatures (SAT) in MERRA-2 are well aligned with station

observations across the GrIS (Reeves Eyre and Zeng, 2017). For these reasons, we selected MERRA-2 to provide (winter) precipitation and SAT data for our analysis.

The Greenland Blocking Index (GBI) used here is defined as the weighted mean 500 hPa geopotential height over a region bounded by 60 – 80 °N and 20 – 80 °W. The dataset is obtained from NOAA Earth System Research Laboratory (ESRL) and

based on Hanna et al. (2013).

### 2.5. GrIS mass balance estimates from GRACE satellite observations

The gravimetric mass balance data used here are from the Greenland Ice Sheet Climate Change Initiative (CCI) project, based on Gravity Recovery and Climate Experiment (GRACE) satellite pair measurements and the GRACE ITSG-Grace2016 gravity field model (Mayer-Gürr et al., 2016). The data covers 2003-2015, as prepared by the Technical University of Dresden (Groh

and Horwath, 2016).

The CCI mass balance estimates are based on a point mass inversion method after Forsberg and Reeh (2007), Sørensen and Forsberg (2010), and Barletta et al. (2013). A notable feature in the use of GRACE for GrIS mass loss estimation is that the gravimetric signals from outer ice caps and small glaciers are not distinguishable from the ice sheet gravity changes. However,

on the scale of the whole ice sheet, the optimal mass balance solution needs to contain this contribution. As the GrIS extent defined for CLARA also contains any outer glacier or ice cap grid cells if their ice coverage is sufficiently large, there is no major discrepancy in the combined use of GRACE and CLARA data.

### 2.6. GrIS surface albedo dataset from MODIS (MOD10A1)

We obtained the latest (Collection 6) MODIS-based GrIS surface albedo data from the MOD10A1 timeseries (Hall et al.,

1995) in order to repeat the analysis by Colgan et al. (2014) and compared these results to the CLARA-A2 SAL-based GrIS surface mass balance proxy. These data are provided in 5 km spatial resolution as daily means between 2000 and 2017, and



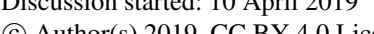


feature denoising, gap filling and bias correction procedures as described in Box et al. (2017). For consistency, these data are also reprojected to the 25 km CLARA grid and grid cells with ice cover less than 50% were discarded.

## 2.7. Methods

### 2.7.1. Study area definitions

As mentioned, here we define GrIS extent through the use of the PROMICE ice mask (Citterio and Ahlstrøm, 2013) using a threshold of 50% for fractional ice cover as the inclusion condition. Basin-scale analyses were done using the six basin boundaries based on Sasgen et al. (2012), with the modification that we combine their basins D and E into our basin 4. The basin delineation and GrIS extent are illustrated in Figure 2.

### 2.7.2. Trend assessment

All trends shown here have been calculated as Theil-Sen estimators, i.e. as the median of all possible slopes between data pairs in the examined dataset (Theil, 1950; Sen, 1968). The Theil-Sen estimator is robust against outlier influence compared to ordinary least squares regression. The trends in surface albedo were deemed significant only if both of the following conditions were met. First, the observed trend slopes were non-zero at the 95% confidence interval. Second, the decadal albedo trends were larger than 0.015 per decade. The latter condition largely follows the analysis by Casey et al. (2017) for the MODIS

Collection 6 GrIS surface albedo, who considered a decadal trend of 0.01 as the limit of MODIS observability. We have tightened that limit to 0.015 for CLARA to account for the less precise radiometric accuracy of AVHRR, which remains even after the considerable intercalibration work done on the CLARA input radiances (Heidinger et al., 2010). We finally note that while the observed trend magnitudes were largely insensitive to the choice of CLARA input data (5-day or monthly means), monthly means are used for the decadal trend calculations when shown here.

### 2.7.3. Melt rate and ice exposure calculations

In order to examine changes in the intensity of melt across the GrIS margins during our 1982-2015 study period, it is necessary to estimate two parameters: the melt rate, expressed as albedo decrease per day during the melting season, and a yes/no condition for determining if a particular area of GrIS melted substantially enough to be composed primarily of bare ice and wet snow. The detection of melt onset over GrIS is challenging, as reductions in albedo typically begin gradually and sporadic

snowfall events may 'reset' the albedo to the pre-melt range. We tested the use of a sophisticated Gaussian Process (GP)-based change point detection algorithm implemented after Fearnhead (2006) and Xuan and Murphy (2007), but the gradual impact of melt onset in the albedo data proved too difficult to capture with such a probability-based change point detection method.

We then implemented a simpler scheme, in which the 5-day albedo record between May-August for each grid cell and year is

first smoothed with a GP regressor algorithm with a Mátern kernel to contain 120 data points, corresponding to a temporal



resolution of ~1 day. The smoothed data series is evaluated to find the first sample after early May for which the albedo value deviates more than two standard deviations from the preceding early-summer albedo mean. The melt rate estimate is then taken to be the albedo difference between that value and the summer minimum albedo, divided by the time between the two events. As this method may be expected to provide reasonable melt rate estimates only for the ice sheet margins where significant
seasonal albedo dynamic exists, we present the melt rates only for areas where the surface elevation is less than 2200 m a.s.l.

This smoothed timeseries was also applied to the basin-scale examination of the relationship between ice discharge rate and albedo changes. Specifically, we estimated the timing of fastest albedo decrease as a proxy for maximum runoff production by finding the day (limited between May and mid-August) in the smoothed, basin-averaged albedo data on which the albedo
decrease rate first reached 95% of its annual maximum. To enhance the albedo dynamic, grid cells with elevations larger than 2800 m were excluded from the aggregation.

For the detection of bare ice and wet snow surfaces, we first note that melting bare ice surfaces around the GrIS margins generally have surface albedos in the range of 0.5 to 0.65 (Bøggild et al., 2010), although significant amounts of impurities at
the ice surface may of course further reduce the albedo. This albedo range is also applicable for wet snow areas with significant melt ponding. We therefore selected an empirically suitable threshold albedo of 0.58 and observed the CLARA grid cells in which the surface albedo reaches this value for each year in the study period. Changes in the melt rate and distribution of bare ice/wet snow areas were studied by comparing three-year means at the beginning and end of the full study period (1982-2015), as well as the MODIS (2000-2015) and pre-MODIS (1982-1999) eras. For the continuous melt rate estimates, trends for these
time periods were further determined using the Theil-Sen estimator as described in section 2.7.2.

### 2.7.4. Forming a surface mass balance proxy from surface albedo observations

The mass balance of GrIS may be expressed as (e.g. van den Broeke et al., 2017)

$$MB = \frac{dM}{dt} = SMB - D \tag{1}$$

Where SMB is the ice sheet's surface mass balance, and D represents the summed solid ice discharge from all large marine-
terminating outlet glaciers. Furthermore, the surface mass balance may be expressed as (e.g. van den Broeke et al., 2017)

$$SMB = P_{tot} - SU_{tot} - ER_{ds} - RU \tag{2}$$

Where $P_{tot}$ is the sum total of snowfall and rainfall, $SU_{tot}$ represents the total sublimation, $ER_{ds}$ represents surface snow mass lost via wind-driven erosion, and RU represents meltwater runoff.

An earlier study by Colgan et al. (2014) presented the idea that, because of the very strong impact of surface melt on the ice sheet's surface mass balance (e.g. van den Broeke et al., 2016), the ice sheet average surface albedo could serve as a meaningful and stationary proxy for the melt season mass balance (MB) dynamic when calibrated against gravimetric mass balance




observations from the Gravity Recovery and Climate Experiment (GRACE) satellite pair. However, the full mass balance of the ice sheet includes dynamic ice losses from glacier discharge, to which surface albedo is insensitive. We therefore propose to expand on the original idea by directly correlating CLARA-observed surface albedo and GRACE-based SMB changes in order to infer a proxy SMB timeseries from the CLARA surface albedo dataset, covering its full 34-year duration and thus

reaching further back in time than either GRACE or a MODIS-based proxy. The method is limited in scope to the summer months (May-August), and is predicated on the dominant influence of runoff in SMB variability during the satellite era (Box, 2013).

To obtain the proxy, we first correct the monthly GrIS MB change record from GRACE data with state-of-the-art discharge

(D) estimates from recent work by King et al. (2018). Then, a linear regression is sought between the new GRACE SMB estimate    and    CLARA    monthly    mean    GrIS    albedo    for    all    summer    months    in the overlap period 2003-2015 for which valid GRACE data exists (N=45). The regression period thus contains data from the period of rapid GrIS mass loss in the 2000s, as well as from the years 2013-2015 when the mass loss had somewhat decelerated. The regression parameters are used to create summed MJJA surface mass balance proxy data from CLARA/GRACE for 1982-

2015 for comparison with corresponding MAR SMB. The variability range observed for central GrIS albedo in CLARA (0.02) is taken to be the uncertainty range of the proxy SMB.

We also calculate the proxy SMB from the most recent MOD10A1 data as described earlier. The obtained relationship is quite similar to that reported by Colgan et al. (2014) for MB, mainly because annual variations in GrIS discharge are relatively

modest compared to the variability in SMB and albedo. The MODIS/GRACE proxy SMB is presented for comparison purposes for the period 2000-2017.

## 3. Results

### 3.1. Greenland surface albedo trends, 1982-2015

The Theil-Sen decadal albedo trend estimates are shown in Figure 3. Here, we omit the years 1992 and 1993 from analysis as

their albedo estimates are less reliable. Over the full CLARA period, we note that GrIS albedo has largely remained stable during the early summer months of May and June, though some of the outermost ice sheet margins in the North and East show statistically significant albedo decrease in June. However, in July and August, we observe significant and negative albedo trends along most of the GrIS margins. Analyzing the CLARA subperiods corresponding the pre-MODIS and MODIS eras (Figure 3, center and bottom rows), we find that the majority of the albedo decrease signal originates after 2000, with a strong

albedo decrease (up to -0.05 / decade as an area average, individual grid cells may reach -0.1 / decade) along the Kangerlussuaq sector on the GrIS west margin. The MODIS era trends are consistent in magnitude and pattern with those presented by Casey et al. (2017) from the latest MOD10A1 time series. The negative trends in CLARA along the northern margin may be





somewhat larger than those in MOD10A1. The cause may be in the tendency of MOD10A1 for retrieving relatively high albedo estimates for the northern parts of GrIS, a feature shown by Alexander et al. (2014) for the previous MODIS collection, which we note to remain valid for the current Collection-6 MOD10A1 for the month of August.

For the rarely examined pre-MODIS era of 1982-1999, we note that while the ice sheet's albedo was primarily stable, significantly negative decadal trends in late-summer albedo (subplots (g, h)) manifest over the Northeast ice sheet. The cause of this albedo decrease remains unclear at present. Based on in situ energy balance measurements over this area during summer of 1994, Konzelmann and Braithwaite (1995) argued that surface melt in the region was primarily driven by net radiative fluxes during a relatively cloudless summer, but also that the relatively high downslope wind speeds increase the importance
of turbulent fluxes, possibly making the area more sensitive to near-surface air temperature anomalies. Mouginot et al. (2015) also reported an increase in modelled runoff in this area from the late 1980s onward, which is consistent with the negative albedo trend seen here.

Because the Theil-Sen trend estimator is defined as the median of all possible slopes between data points in a time series, the
subperiod trends shown in Figure 3 do not necessarily sum up to the trends over the full 1982-2015 period. However, because the albedo trends from MODIS time series data have been extensively studied, we felt that presentation of the subperiod trends is important to compare results.

The subperiod division used here is also useful in that the clear change in late-summer albedo trends between 1982-1999 and
2000-2015 is fully consistent with a documented change in atmospheric regime over GrIS which began to favour high-pressure blocking patterns over GrIS, as well as northward transport of warm and moist air, increasing near-surface air temperatures and enhancing downward longwave energy fluxes, all of which are features which promote surface melt (e.g. Box et al., 2012, Hanna et al. (2013), Mioduszewski et al. (2016), Välisuo et al. (2018)). Some of the other potential causes of the recent darkening of the western margin include increasing biological activity (algae growth; Ryan et al., 2018) and previously
deposited impurities emerging from the ice with each melting season, although recent field measurements have not detected evidence of this (Polashenski et al., 2015).

We finally observe that some of the outermost grid cells of e.g. the western edge of the ice sheet display very small negative or even positive trends for all studied months. This finding, while somewhat surprising, is consistent with earlier field
observations stating that the outermost zone of the ice sheet margin consists of clean ice, effectively drained of melt water by ubiquitous crevasses and moulins (Knap and Oerlemans, 1996) and containing fewer biotic or abiotic impurities, and thus resistant to albedo decreases caused by enhanced melting. However, we note that with the relatively coarse spatial resolution of CLARA, the outermost grid cells are mixed pixels even after the exclusion of grid cells with <50% ice cover. The albedo



of these pixels includes relatively constant values of the bare soil surrounding the ice, which could also explain the observed stability. In any event, the impact of these areas on the overall GrIS albedo trends is negligibly small.

### 3.2. Melt intensity and bare ice exposure

The left column (subplots (a, d, h)) in Figure 4 shows the change in the mean albedo decrease rate, taken as a proxy for melt season intensity, between the last and first 3 years of each period. Here, the decrease rates are shown only for elevations below 2200 m a.s.l., where they are expected to be more robust. The overlaid pentagons (and crosses) indicate areas where albedos typical of bare ice and wet snow have been reached at the end but not the beginning of each period (and vice versa). The patterns correlate well spatially, as expected. We see increases in melt intensity along the western margin (Basin 5) for the full

CLARA period and both subperiods, but the MODIS era has seen systematically faster albedo decrease rates also over Basin 6 further north. To assess the consistency and robustness of the changes in melt intensity, we also calculated the Theil-Sen decadal trend estimates for the melt rates during each period, shown in Figure 5. The acceleration in melt rate between 1982 and 2015 is significant over large areas in the western, northern, and northeastern GrIS margins, largely as a consequence of a changes seen during 2000-2012.

Also, while some areas saw new bare ice exposure in the late 1990s (relative to early 1980s), there has been a widespread increase in areas experiencing bare ice exposure (or very wet snow cover) during the 2000s, so that these areas now routinely reach elevations 50 – 100 m higher up the ice sheet compared to 1982-1984 (Figure 4a, inset). These changes are consistent with corresponding positive anomalies in MERRA-2 mean SAT during the summer months, as shown in Figure 4, subplots

(c,f,j).

A notable and interesting deviation from this trend towards faster melt and expanding bare ice / very wet snow cover has occurred along the southeastern margin, roughly between Tasiilaq and Paamiut (Basin 4). Here, we observe no increase in melt rate during the MODIS era, and areas which reached bare ice albedos during the melting season at the turn of the century

no longer do so in 2013-2015. We suggest that both of these effects are explained by substantial increases in winter snowfall over this region, as shown by the MERRA-2 winter precipitation anomalies (Figure 4, subplots (b,e,i)). The increasing snow deposition creates a buffer against melt rate acceleration, inhibiting bare ice exposure at the same time. The summer SAT anomalies during the MODIS era over this region are also less significant than those over the western margin, consistent with prior analysis suggesting that the presently dominant atmospheric circulation patterns particularly favour northward warm air

transport along the western margin of the ice sheet (Fettweis et al., 2013), potentially enhanced at times by the so-called 'atmospheric river' events (Neff et al., 2014).



### 3.3. GrIS Surface Mass Balance (SMB) changes in summer during the past three decades

Figure 6A shows the Theil-Sen regression fit between the monthly means of the CLARA albedo timeseries and the GrIS SMB, obtained by correcting the GRACE mass balance observations with the state-of-the-art monthly ice discharge estimates (King et al., 2018) after eq. (1). The obtained fit shows no temporal dependency (years marked by colors in Fig. 5a), is fairly robust ($r^2 \sim 0.77$), and is of the form

$$SMB = 3028 \times \alpha_{CLARA-BSA} - 2222 \tag{3}$$

This sensitivity of SMB to albedo changes in this relationship, which we emphasize to be a statistical one, is markedly different between the CLARA-GRACE pair relative to the MOD10A1-GRACE pair, whose fit is less robust ($r2 \sim 0.45$) and reproduced here:

$$SMB = 1711 \times \alpha_{MOD10A1-BSA} - 1378 \tag{4}$$

Most of the difference is likely attributable to a substantially larger August mean GrIS albedo in MOD10A1 relative to CLARA. The MOD10A1 mean GrIS albedo is generally higher than in CLARA for any given month but this difference is most pronounced in August, likely due to different retrieval algorithms between the two products.

When applying the relationships of eq. (3) and eq. (4) to the respective MJJA albedo timeseries and summing to the summer SMB estimates, we obtain the results shown in Figure 6b. The original GRACE SMB estimates and the calculated MAR SMB are shown for comparison. Both proxy SMB timeseries overestimate MAR SMB, largely because the original GRACE SMB estimates do so as well. However, it is notable that the GRACE proxy SMB largely does agree with MAR SMB between 1982 and 2014 within its uncertainty envelope, apart from the a priori known period of 1992-1993, and 1999-2000 to a smaller extent. The 1999-2000 disagreement may be traceable to a change in the NOAA AVHRR constellation which occurred at the time. Both CLARA and MOD10A1 proxy SMB exhibit a significant Pearson correlation coefficient against MAR (CLARA: 0.79, MOD10A1: 0.95), indicating that the proxies capture most of the annual variability in summer SMB.

A notable divergence between the proxies and MAR in the MODIS era occurs in 2012, when evidently the summer mass loss clearly outpaced any albedo reductions seen by either MODIS or AVHRR. A potential explanation is that albedo decreases are physically bounded in a way that the surface-melt induced mass loss is not. The albedo of the ice sheet margin, where the changes are concentrated, could be considered to have physically based local effective minima, corresponding to conditions where the ice sheet surface is consistently melting and saturated with melt ponds and melt rivers up to the local effective maximum coverage. In this logic, beyond this point the mean surface albedo of the area would have almost nowhere to go in terms of further decreases, barring the introduction of large concentrations of light-absorbing impurities or biomass.





Considering the completely independent nature of the proxy record sources in relation to each other (optical imagers vs. GRACE) on one hand, and between proxy observations and MAR model output on the other, the agreement seen in Figure 6B serves to first confirm the similarity of albedo reductions seen in MOD10A1 and CLARA, and also to implicitly confirm the dominant nature of runoff increase in the post-2000 rapid surface mass loss era (e.g. van den Broeke et al., 2016) on the other,

as the albedo-based proxies can only be considered to be well-correlated to runoff in eq. (2). The general agreement of the CLARA proxy record with MAR within the AVHRR observational uncertainty also serves as an observation-driven confirmation of the stability of SMB in the 1982-1999 period at the scale of the ice sheet.

### 3.4.    On the basin-scale relationships between surface albedo as a runoff proxy, and ice discharge

A comparison between detrended, normalized basin-scale D anomalies and surface albedo (serving as a proxy for runoff)
between 2000 and 2015 yielded no statistically significant relationships for any examined summer month at the 95% confidence level. This negative finding implies that surface runoff anomalies do not significantly alter the solid ice discharge rates through the GrIS outlet glaciers at the annual level. This is in line with previous modelling work (Schoof, 2010, Tedstone et al., 2015) which has suggested a complex relationship between meltwater availability and glacier flow rate, where large amounts of meltwater may serve to decelerate rather than accelerate glacier basal flow. Also, King et al. (2018) note that the
annual variability in GrIS ice discharge is mostly connected to cumulative changes in the calving front positions of its outlet glaciers.

However, Schoof (2010) as well as Tedstone et al. (2015) and King et al. (2018) note that short-term increases in meltwater runoff can temporarily increase basal water pressure, leading to glacier acceleration. This relationship is active in the early
part of the melting season and typically operates at a narrow temporal window, with the current estimate for the temporal lag between fastest increase in runoff and the maximum GrIS D being $13 \pm 9$ days (King et al 2018). To examine this connection from observational evidence, we again adopted the surface albedo changes in CLARA-A2 SAL as a runoff proxy, and compare the timing of the maximum (detrended and normalized) D in each basin with the timing of fastest basin-scale albedo decrease, following the calculation outlined in section 2.7.3.

The thus-obtained time lags (runoff proxy maximum minus D maximum) are shown in Figure 7. While year-to-year variability remains high, most of the years at most of the basins exhibit time lags within or similar to the expected time window. Temporal differences between maxima are largest in Basin 1, where there is generally more gradual sloping bed topography. By contrast, we note that temporal differences are smallest in the southeastern Basin 4, where glacier bed topographies are much steeper.
The presence of negative time lags in the comparison also remind us that uncertainties still exist in the analysis, particularly in the determination of maximum runoff date from relatively noisy surface albedo data, even after statistical smoothing operations. Nevertheless, the close temporal proximity between maximum discharge and our observation-based runoff proxy





in most basins suggest these two processes are dynamically related, supporting previous findings that runoff can influence glacier flow by altering subglacier water pressure and sliding speeds.

## 4. Discussion

The CLARA-A2 SAL surface albedo estimates are not normalized to any particular Sun Zenith Angle (SZA), but rather represent the per-grid cell mean solar illumination conditions of each 5-day or monthly time period. This choice is consciously made, as typically used SZA normalization algorithms (Briegleb et al., 1986; Gardner and Sharp, 2010) assume a noon-symmetrical 'u-shape' for the diurnal cycle of snow albedo. However, several in situ measurement campaigns have shown that this choice is not valid for snow during the melting period, when the 'u-shape' is often noon-asymmetrical (Dirmhirn and Eaton, 1975; Jonsell et al., 2003; Pirazzini et al., 2006; Meinander et al., 2013). As here we study the GrIS margins, where annual melting seasons are intensive, we have chosen not to present primary results based on an uncertain normalization procedure. For completeness, Supplementary Figure S1 reproduces Figure 3 with the application of the SZA normalization proposed by Gardner and Sharp (2010) with a target SZA of 60 degrees. The trends are consistent, demonstrating that the results shown here are not sensitive to the inclusion or exclusion of SZA normalization. Also, the separate treatment for each summer month and the use of a conservative SZA cutoff (of 70 degrees) in the albedo estimation will further ameliorate any effects of this choice.

Regarding the recent inhibition of surface melt along the SE margin of the ice sheet, which we suggest is due to increasing winter precipitation, there are somewhat conflicting views in the recent literature. Berdahl et al. (2018) examined observation precipitation records at Tasiilaq station, taken to be representative of the surrounding parts of the SE ice sheet, and found no significant December-February precipitation increases in the satellite era up to 2012. However, Koyama and Stroeve (2018) evaluated the Arctic System Reanalysis precipitation against in situ observation along the GrIS margins, and it is notable that in situ observations in their study indicated increasing precipitation in the 2010-2015 period at sites 04390 and 04272, near the southern tip of Greenland. These increases are in line with the increasing winter precipitation signal we observe in MERRA-2, and consistent with the lack of new albedo reductions or bare ice exposure in CLARA (Figure 4). The curiously bimodal structure of the MERRA-2 winter precipitation anomalies between 2000-2015, with increases in the southernmost parts of the SE margins and reductions towards the east, is noteworthy but its deeper analysis is beyond the scope of this study. It is intriguing to note, however, that Tasiilaq station appears to lie at the no-change zone between the MERRA-2 bimodal peaks, which might serve to explain the lack of observed precipitation increases by Berdahl et al. (2018). Lastly, the efficacy of the melt inhibition depends fully on the phase of the precipitation; recent evidence pointing towards an increasing replacement of snowfall with rainfall over southern GrIS may prove our finding to be transient in nature (Oltmanns et al., 2019).

The general agreement between modelled and observation-based SMB shown here contains substantial (though mutually cancelling) disagreements at the monthly level. Most notably, the ERA-Interim-forced MAR v.3.5.2. SMB generally sees the





largest annual mass loss occurring in July, whereas the corresponding GRACE SMB estimates assign the largest mass loss into the month of August. Such differences have emerged in prior studies (Alexander et al., 2016), although based on different GRACE mass balance solutions and a MAR version of different forcing and spatial resolution, invalidating direct comparisons between results. A part of the reasonably good covariability between MAR SMB and albedo-based proxy SMB likely also lies

in the better agreement between GRACE and MAR mass balance at elevations below 2000 m a.s.l. (Alexander et al., 2016), where also most of the GrIS albedo dynamics occur.

The ESA Greenland Ice Sheet CCI distributes mass balance solutions from two algorithms (TUDresden and DTU). For completeness, we repeated the SMB analysis also for the DTU GRACE data. The results (Supplementary Figure S2) show a

less favourable fit between CLARA and discharge-corrected GRACE SMB with a larger slope, leading to generally much less negative surface mass balance in the CLARA-GRACE proxy relative to MAR. The MOD10A1 is similarly affected but to a lesser degree. It is notable that both discharge-corrected GRACE SMB timeseries from ESA Greenland Ice Sheet CCI produce systematically less negative SMB during the summers 2004-2010 relative to MAR.

A variety of drivers of the post-2000 increases in GrIS surface melt (and thus decreases in surface albedo) have been proposed. First are the atmospheric circulation pattern changes, which have led to warm and moist air advection over the ice sheet (e.g. Fettweis et al., 2013; Mioduszewski et al., 2016) leading to both increases in SAT (Reeves Eyre and Zeng, 2018) and increases in clear-sky downwelling longwave radiation (Mioduszewski et al., 2016; Välisuo et al., 2018). Another viewpoint has been that changes in the surface radiative energy budget, caused by cloud coverage and radiative property changes, are the primary

driver. Hofer et al. (2017) argued, using the CLARA-A1 dataset, the predecessor of CLARA-A2 used here, that decreasing cloudiness over the GrIS ablation zone in the 2000s has caused a significant increase in insolation. This effect has combined with the increasing clear-sky downwelling longwave radiation from the warm air mass advection to cause the increasing melt. Along this theme but based on a different mechanism, Bennartz et al. (2013) argued that over the high-elevation center of the ice sheet, intrusions of optically thin, low-level, and liquid-bearing clouds drive the downward longwave flux up significantly,

yet are thin enough to allow shortwave flux penetration to the surface, promoting extensive surface melt.

Here, we have focused on analysing the surface albedo changes using reanalysis SAT and winter precipitation as supporting evidence. For comparison purposes, we calculated the mean MJJA cloud fractional cover over GrIS from CLARA-A2 (1982-2015) to compare against the corresponding MAR cloud cover field. The results in Figure 8 show the change in CLARA cloud

cover resulting from updated algorithms and AVHRR radiance calibration, resulting in a recovery of cloud cover after 2007 whereas MAR cloud cover continued to decline until 2011. This highlights the uncertainties in cloud observations from legacy AVHRR data, even though the detection performance has been shown to improve in CLARA-A2 (Karlsson et al., 2017), with a limited study over central GrIS during 2010 showing good agreement between CLARA and MODIS cloud cover (Riihelä et



al., 2017), implying that the latest years of CLARA with the highest number of concurrently serving AVHRR instruments should be the most reliable.

While the overall cloudiness decline from the mid-1990s to 2007 is still in line with the melt mechanisms proposed by Hofer et al. (2017), the post-2007 cloudiness recovery in CLARA-A2 is then inconsistent with the continuing large mass loss from the CLARA and MOD10A1 proxy SMB timeseries, as well as MAR, up till 2012. This suggests that the relationship between cloudiness, atmospheric circulation patterns, and surface melt over GrIS may be still more complicated than has been shown to date. Another complication lies in that not only the extent of cloud cover, but also its radiative properties, be they expressed through Cloud Optical Thickness (COT) and effective particle radius, or Liquid Water Path (LWP), are very important for correctly estimating the surface radiative energy budget over bright snow – and these retrievals from AVHRR are challenging (Riihelä et al., 2017).

Another means of linking large-scale atmospheric circulation to the surface albedo changes is via the Greenland Blocking Index (GBI; Hanna et al., 2013, 2016). Positive anomalies in GBI represent atmospheric conditions which block northerly zonal air flow over the ice sheet, allow for warm southerly air flow particularly across the west GrIS margin, and promote clear sky conditions over the ice sheet, enhancing surface melt (Hanna et al., 2014). The basin-scale comparisons between surface albedo anomalies and GBI anomalies (both calculated with respect to the 1982-2015 mean) are shown as time series in Figure 9.

The negative correlation between GBI and albedo is evident (scatterplots available as Supplementary Figure S3), although different basins display different responses. For example, in agreement with Tedesco et al. (2016), the 2015 positive GBI anomaly was linked to an atmospheric ridge which enhanced melt in northern Greenland (basins 1 and 6 while preventing it in the south, a feature reproduced in the corresponding albedo anomalies. Along with the concurrent recovery in GrIS cloudiness (Figure 8), the significant covariability between GBI and albedo implies that the apparent post-2012 slowing in further albedo decreases is linked to a shift in large-scale atmospheric circulation, allowing for more cold northerly airflow (Tedesco et al., 2014) and increasing cloudiness (during summers).

Finally, we note that while the CLARA proxy offers a new look into period for which observation-based and spatially or temporally comprehensive estimates for SMB have been previously unavailable, the method shown has its limits. The SMB proxy may be derived only for the summer months when albedo retrievals are possible. Radiometric calibration and cloud detection uncertainties propagate into the surface albedo estimates as unwanted noise, which we have attempted to account for through an uncertainty envelope derived from a stability examination. Also, the results show clear physical constraints on the validity of mechanisms linking enhanced albedo reductions as a proxy for SMB decreases. The question of increasing biological activity as a driver of albedo reduction along the GrIS margins remains open and no long-term records exist for



comparison with the present albedo estimates from CLARA. A large-scale spread of lichen colonies on the ice surface would certainly provide a mechanism for albedo decrease unaccounted for in present models and satellite-based retrieval algorithms.

## 5.    Conclusions

5    We have investigated the decadal trends in the surface albedo of the Greenland Ice Sheet between 1982 and 2015, and their connection to melt rate, bare ice exposure, as well as surface mass balance and ice discharge rate. The main results of this study may be summarized as follows.

- Over the 34-year investigation period, the late-summer surface albedo of the GrIS margins shows statistically significant decrease at the 95% confidence level. The 1982-2015 decadal albedo trends reach a maximum of approximately -0.05 over the outermost ablation region in the Kangerlussuaq sector in July, although some grid cells in the area display decadal trends twice as large during 2000-2015.

- The albedo decrease of the northeastern and eastern margins was initiated during the 1982-1999 period, whereas the decrease along the west coast mainly occurred between 2000 and 2012, concurrently and consistently with a previously reported change in atmospheric circulation favouring the advection of warm and moist southerly airmasses along the west coast.

- The melt seasons along the GrIS intensified after the turn of the millennium, as seen by an acceleration of the albedo decrease rate during the melting season. Also, regions where surface melt causes the albedo to drop to values typical of bare ice were much more widespread in 2012-2015 than in the early 1980s, and expand to elevations 50-100 m higher on the ice sheet.

- A notable exception was observed along the southern margins, where we propose that increased winter snowfall has created a buffer against snowmelt intensification and bare ice exposure. The future persistence of this effect will undoubtedly depend on the future magnitude and phase of precipitation along the GrIS southern margin.

- When calibrated against surface mass balance estimates based on GRACE gravimetric measurements and state-of-the-art ice discharge estimates, the CLARA surface albedo estimates form a proxy for SMB reaching back to 1982. The summer-only SMB proxy shows reasonably good agreement and covariability with latest MAR regional climate model estimates for the full investigation period. This indirectly confirms the dominant role of surface runoff in GrIS SMB variability during the satellite era, and highlights the sensitivity of albedo to surface runoff.

- We find no observational evidence for a connection between annual surface albedo changes (as a proxy for runoff production anomalies) and anomalies in annual rates of ice discharge at GrIS drainage basin-scales. This is in agreement with recent modelling results which suggest that annual ice discharge is more sensitive to long-term variability in glacier front positions. However, at shorter timescales, we find that the timings of maximum runoff production, estimated here as the period of fastest albedo decrease, and the seasonal maximum in normalized ice





discharge generally exhibit time lags consistent with a previously reported range of $13 \pm 9$ days. These observations support a surface melt-induced sliding mechanism, where surface melt waters that flow to the ice-bed interface near the margins can temporarily increase basal water pressures and glacier sliding speeds.

5 Overall, the results fully fit into the present picture of GrIS being an ice sheet under duress, particularly between the mid-1990s and 2012. The decreasing albedo along the margins promotes enhanced surface melt, although atmospheric circulation patterns, cloud cover frequency and properties, as well as precipitation intensity and phase are most likely the strongest drivers of change manifested through surface mass loss and peripheral darkening of the ice sheet. Years 2013-2015 show amelioration in the albedo decreases over GrIS margins, yet the longevity of this feature remains to be seen.

10 **Data availability**

The CLARA-A2 surface albedo data record is available from https://doi.org/10.5676/EUM_SAF_CM/CLARA_AVHRR/V002. The MAR climate model data was obtained from ftp://ftp.climato.be/fettweis/MARv3.5/Greenland. The Greenland Ice Sheet CCI GRACE data was obtained from http://products.esa-icesheets-cci.org/products/downloadlist/GMB/. The Greenland Blocking Index was obtained from 15 https://www.esrl.noaa.gov/psd/data/timeseries/daily/GBI/.

**Author contribution**

A.R. planned the study and performed the analyses. M.K. was in charge of the discharge estimate data and contributed to the manuscript text. K.A. was the responsible scientist for the creation of the CLARA-A2 SAL dataset and contributed to the manuscript text. A.R. prepared the manuscript with contributions from all co-authors.

20 **Competing interests**

The authors declare that they have no conflict of interest.

**Acknowledgments**

The work of A.R. has been funded by the Academy of Finland, decision # 287399. Contributions from M.K. were supported by grants 80NSSC18K1027 and NN13AI21A from the US National Aeronautics and Space Administration.



The CLARA-A2 SAL data record was created by the Satellite Application Facility on Climate Monitoring (CM SAF), a project of EUMETSAT. Colleagues in the CM SAF project are thanked for their support in the creation of the CLARA-A2 SAL dataset. Supplementary Figures S1-S3 are available in the supporting information of this manuscript.

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

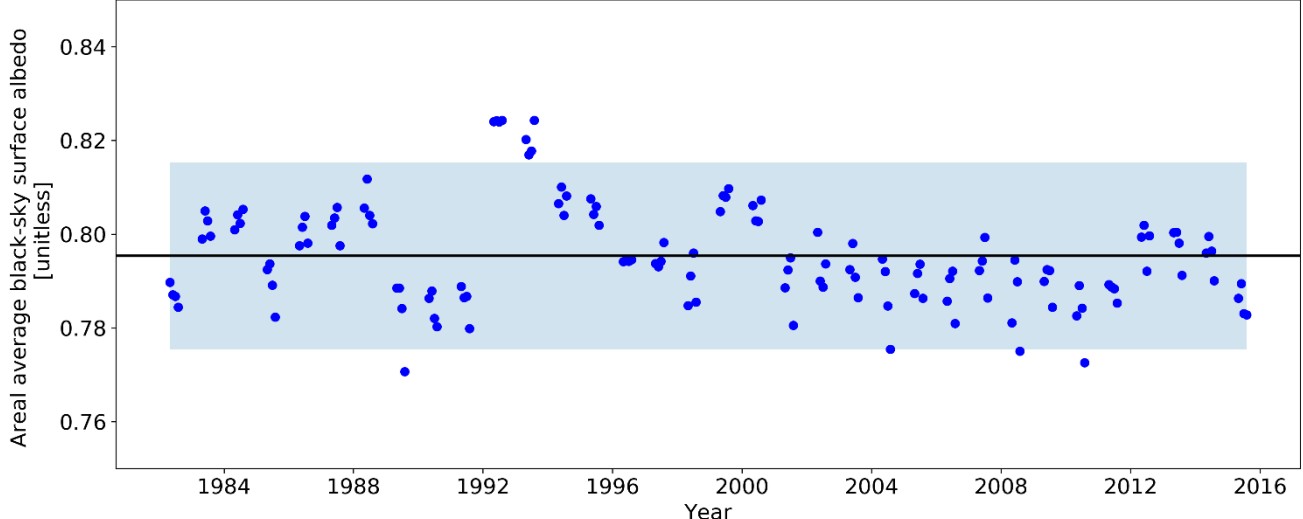

**Figure 1: Stability of CLARA-A2 SAL over 73-75 °N, 38-42 °W, on the central part of GrIS. Spatially averaged monthly mean albedo retrievals between May and August shown. The black line shows the 34-yr mean albedo of the area. Light blue shading shows ±0.02 albedo variability about the interannual mean, corresponding to the reported upper accuracy estimate of 5% relative for CLARA-A2 SAL over snow and ice.**

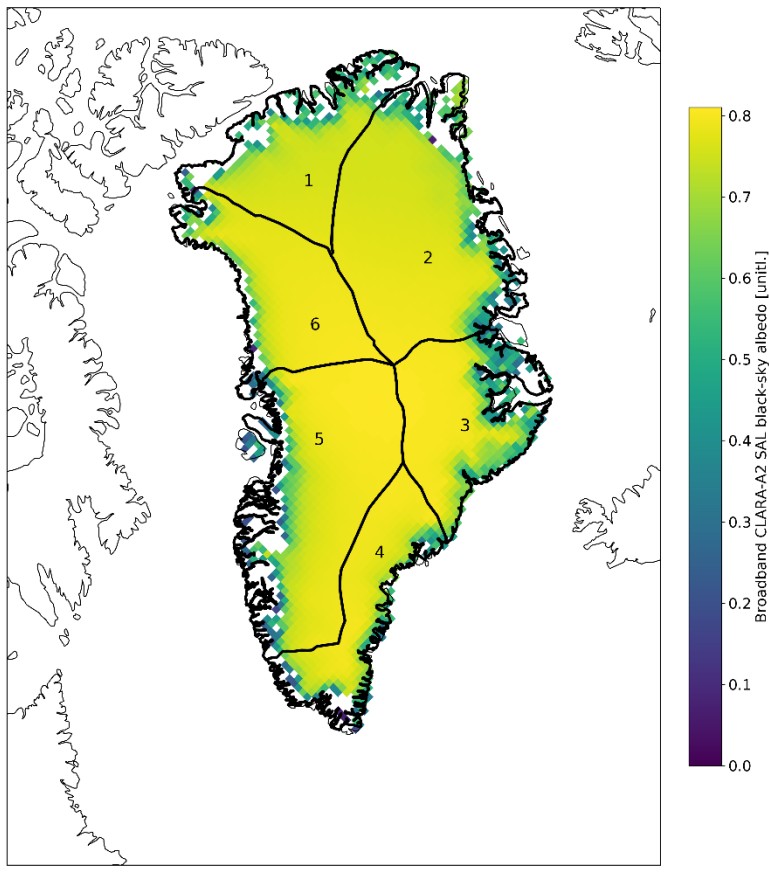

**Figure 2: GrIS drainage basins as used in this study. Basin boundaries overlaid on an example of CLARA-A2 SAL monthly mean surface albedo from June 2010.**





**Figure 3: Theil-Sen decadal albedo trends over GrIS from CLARA-A2 SAL during 1982-2015 (a-d), 1982-1999 (e-i), and 2000-2015 (j-m). Columns correspond to the summer months May-August from left to right. The hatched regions in the trend figures indicate areas over which the trend is significant (i.e. non-zero slope) at the 95% confidence level, and its magnitude exceeds the uncertainty limit of 0.015/decade. Years 1992 and 1993 are excluded. Color bar limited to highlight majority values.**







**Figure 4: Subplots (a,d,h): Estimated mean albedo decrease rate changes between the last three and first three years of each analysed period. Green-yellow pentagons indicate areas where bare ice albedo is reached during the last but not the first years of the period. Crosses indicate vice versa. Subplots (b,e,i): MERRA-2 mean winter precipitation anomaly [m w.e.] of each period vs. 1982-1992 mean. Subplots (c,f,j): MERRA-2 mean May-August SAT anomaly of each period vs. 1982-1999 mean. Period rows ordered as in**

5 **Figure 3.**

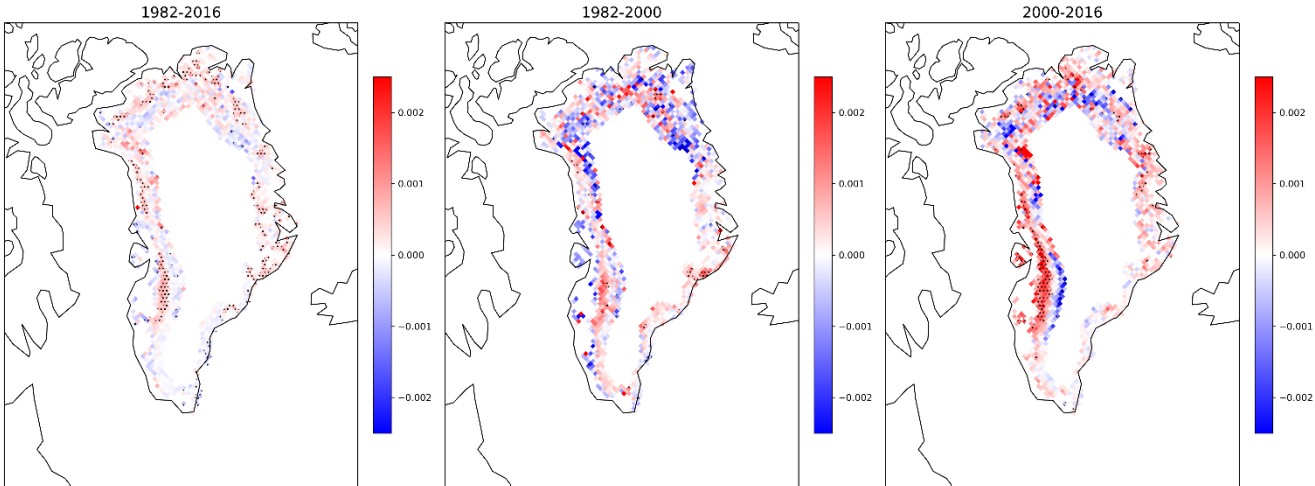

**Figure 5: Theil-Sen decadal trend estimates in albedo decrease rate [units per day] serving as a proxy for melt intensity. Hatched pattern indicates areas where the trend is statistically significant (non-zero slope) at the 95% confidence level. Estimates shown only for elevations lower than 2200 m a.s.l.**



**Figure 6: (a) Linear regression fit between CLARA-A2 SAL monthly mean GrIS surface albedo (x-axis) and the TUDresden GRACE RL06 monthly SMB, obtained by correcting the GRACE mass balance data with ice discharge estimates by King et al. (2018). (b) GrIS May-August (MJJA) surface mass balance sums between 1982 and 2017. Blue line and markers show the CLARA-proxy SMB as calibrated against GRACE observations, with uncertainty envelope (blue shading) based on best-estimate for GrIS mean albedo stability of 0.02 in CLARA. Green line shows the corresponding SMB estimates from the MAR regional climate model. The magenta line and markers show the discharge-corrected GRACE SMB estimate, shown only for years with full GRACE MJJA availability. The red dashed line and markers show the MOD10A1-based SMB estimate for comparison.**





**Figure 7: Basin-averaged annual time lags between maximum runoff production, obtained by proxy from each basin's surface albedo timeseries, and the annual maximum in solid ice discharge (D), obtained from normalized basin-scale discharge rates by King et al. (2018). Blue shaded region illustrates the expected time lag window of 13 ± 9 days from modelling (King et al., 2018). Light blue shading represents the additional temporal uncertainty related to the five-day temporal resolution in the source data. Red (and pink) markers indicate years in which the time lag agrees with the expected range (or with the additional uncertainty taken into account). Note that basins are ordered here by their relative geographical position.**



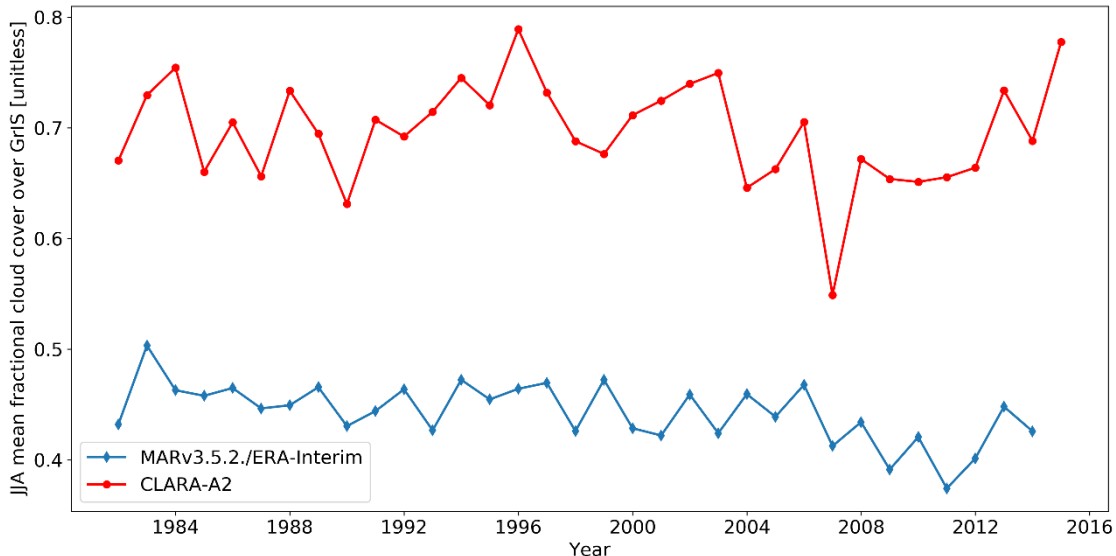

**Figure 8: Mean JJA fractional cloud cover over GrIS from CLARA-A2 and MAR v3.5.2 with ERA-Interim forcing.**



**Figure 9: Comparison of monthly basin-scale CLARA-A2 SAL anomalies (red circle) against Greenland Blocking Index (GBI) anomalies (blue triangle). All anomalies calculated with respect to the 1982-2015 mean albedo or GBI. Red and blue lines show 2-year running means of the respective anomalies. Basins ordered as in Figure 7. Years 1992 and 1993 excluded from the analysis.**

