# Peer review of "The surface albedo of the Greenland Ice Sheet between 1982 and 2015, and its relationship to the ice sheet's surface mass balance and ice discharge"

_The Cryosphere, 2019_

## Referee Comment (RC1) · Anonymous Referee #1 · 6 May 2019

This paper reconstructs the recent SMB decrease by using monthly albedo from a "new" satellite product (CLARA). The authors also try to correlate albedo with ice discharge. It is interesting but, however, I don't recommend to accept this paper in its present state because:

- albedo can be only used, scientifically speaking, as a proxy of melt extent and melt amount. Not as proxy of ice discharge (as shown by the authors) and not as proxy of summer SMB (also in part driven by summer snowfall anomalies, as also mentioned by the authors). Due to the delay between production of meltwater (highly correlated with albedo I agree) and runoff (depending of the snowpack meltwater retention), runoff

can not be correlated with albedo at the monthly time scale. Using MAR outputs, at the summer time scale, it is true that MJJA SMB is correlated at -0.96 (-0.88 over 1980-1999) with runoff and 0.66 with snowfall over 1980-2015. But in May for example, SMB is correlated at 0.98 with snowfall and -0.13 with runoff, showing well that albedo can not be used as proxy of monthly SMB, althrough MAR is not "the true".

- the role of the albedo decrease and the bare ice expansion to the recent melt increase has already been shown a lot of time in previous publications (e.g. Box et al., 2013).

- the correlation between GBI and melt (approximated with albedo here) is also something which is known from a long time (see the Hanna et al., ….).

- the discussion about the cloud cover seem to be out of the purpose of this paper... and using MAR for this is certainly not a robust basis of validation.

To conclude, the authors try to correlate their "new" data with several previous studies/estimates but there is no new interesting scientific message in this paper deserving to accept this paper in TC. However, using this new satellite product deserves to be published, but I recommend to the authors to limit their correlations/comparisons to melt extent (from satellite) or modelled melt amount (as model validation data set). A comparison with the albedo/bare ice extent MODIS based product (notably used as bare ice albedo in RACMO/HIRHAM) will also be more interesting.

---

## Referee Comment (RC2) · Jason Box (Referee) · 31 May 2019

notes on comments: a.) io = "instead of" b.) Comments refer to page then line number using X Y

General Remark The study has a lot of value by presenting an expanded (in time and process) treatment of AVHRR albedo over Greenland. The article makes several major analyses, the fourth of which I am not sure should be kept because of its very wide scope, complexity and limited finding.

Major critique A.) The fourth major analysis should be more clearly explained or re-

moved*, that with lag analysis, basin scale examination of hypothetical meltwater lubrication of ice dynamics. The study already has a lot of substance. Adding the lagged result only to confirm earlier studies is a bit much. 12 9-16 reinforces that the study is taking the empirical albedo relation too far. 1 24-27 recommend removing this part of the study as it does not directly examine melt-induced flow acceleration while much has been evaluated more directly on this topic. See SWIPA 2017 chapter 6 https://www.amap.no/documents/doc/snow-water-ice-and-permafrost-in-the-arctic-swipa-2017/1610 *7 6-11 difficult to follow

B.) Discussion of this study vs Stroeve (2001) Stroeve, J.: Assessment of Greenland albedo variability from the advanced very high resolution radiometer Polar Pathfinder data set, J. Geophys. Res.-Atmos., 106, 33989–34006, doi:10.1029/2001jd900072, 2001. How does this study square with Stroeve (2001, see Fig 4 etc) who found decreasing trends 1981-1998? 8 29 "majority of the albedo decrease signal originates after 2000" but Stroeve 2001 found a decrease before 2000 9 19-26 bringing in Stroeve 2001 agreement/disagreement seems important here Stroeve found NAO resonance, as one would expect. What about this study?

C.) conclusions. . . 16 12 "The albedo decrease of the northeastern and eastern margins was initiated during the 1982-1999 period". . . you offer a mechanism for the west but what about the east...any idea the cause? It should be either/and atmospheric circulation or sea ice -related.

D.) A direct comparison between albedo data sets: CLARA-A2-SAL and MODIS MOD10A1 seems warranted. How well do they agree in the overlapping period?

General comments 7 16 agree with "empirically suitable threshold albedo of 0.58". . . compositing with many PROMICE years yields 0.56 (unpublished) Recommend to not use abbreviation "GrIS". Instead, use "Greenland ice sheet" until it (very quickly) becomes obvious the study is on Greenland, afterward, use "ice sheet". Should title have "mass balance" io "surface mass balance and ice discharge" 1 9 "driven in part by " io

"primarily driven by " 1 13 "We then subtract ice discharge from the mass change estimates from the GRACE satellite mission to estimate surface mass balance" io "We then correct the mass balance estimates observed by the GRACE satellite mission with state-of-the-art ice discharge" 1 23 "rapid surface mass" io "rapid mass" 2 31 "examining the role of albedo" io "both highlighting and confirming the dominant role of surface melt" would seem to improve the statement by making it not a conclusion placed in the intro of the paper and otherwise clarifying that albedo is the predictor variable here. 4 1 "uppermost areas" io "innermost parts" 4 1 - 4 8 . . .Fig 3A in Box, J.E., D. van As, K. Steffen, 2017. Greenland, Canadian and Icelandic land ice albedo grids (2000-2016), Geological Survey of Denmark and Greenland Bulletin, 38, 53-56 available from http://www.geus.dk/DK/publications/geol-survey-dk-gl-bull/38/Documents/nr38_p53-56.pdf supports the idea that 2012 was not anomalously low AND that variability is small (in the Summit GC-Net example; max-min = 0.03) in the dry snow area' 4 12 "∼0.02 increase of the GrIS albedo" io "∼0.02 overestimation of the GrIS albedo" . . . it was a real climate event so the measurement is not an overestimation 8 25 I expect some readers/reviewers will dislike excluding 92 and 93. Yet, I think it's not too questionable as long as you're clear. Here, better I think would be "externally forced" io "less reliable" 8 25 "largely remained stable" discuss relative to Stroeve 2001 8 31 "may be"? Seems more testing needed to address this hypothesis. 9 1 "is" io "may be". . . see/cite Box, J.E., D. van As, K. Steffen, 2017. Greenland, Canadian and Icelandic land ice albedo grids (2000-2016), Geological Survey of Denmark and Greenland Bulletin, 38, 53-56 9 5 "rarely examined" "the ice sheet's albedo was primarily stable" see Fig 9c and related discussion in the following where from 1988-1999 eastern Greenland has the largest AVHRR albedo decrease. Some discussion of that seems warranted. Box, J.E., D.H. Bromwich, B.A. Veenhuis, L-S Bai, J.C. Stroeve, J.C. Rogers, K. Steffen, T. Haran, S-H Wang, 2006: Greenland ice sheet surface mass balance variability (1988-2004) from calibrated Polar MM5 output, Journal of Climate, Vol. 19(12), 2783–2800. 9 28-34 geolocation errors were attributed in the following study for the relatively noisy ice margin trends. See Box, J.E., D.H. Bromwich, B.A.

[Figure]

Veenhuis, L-S Bai, J.C. Stroeve, J.C. Rogers, K. Steffen, T. Haran, S-H Wang, 2006: Greenland ice sheet surface mass balance variability (1988-2004) from calibrated Polar MM5 output, Journal of Climate, Vol. 19(12), 2783–2800. 10 7 "where the trend signal originates" io "where they are expected to be more robust" 10 10 "larger" io "faster" 10 21 "earlier" io "faster" 10 25 the "increases in winter snowfall" finding is very interesting AND is related to the GRACE correlation because when there is snowfall, mass is added and albedo increases. So, be sure to make that point. The following may be relevant if you want to discuss more about how increasing snowfall may be from climate change. https://iopscience.iop.org/article/10.1088/1748-9326/10/11/114008/meta Further Box et al. (2013) find a climate change signal, an increase in snowfall with NH Air T, N Atlantic Air T, etc. Comparison of Greenland accumulation history with northern hemisphere air temperatures suggests a 6.8% (or 51 Gt) per degree C climate sensitivity (Box et al., 2013). See Box, J. E. 2013. Greenland ice sheet mass balance reconstruction. Part II: Surface mass balance (1840-2010), Journal of Climate,Vol. 26, No. 18. 6974-6989. doi:10.1175/JCLI-D-12-00518.1 15 20-26 Including discussion of Rajewicz and Marshall, 2014; McLeod and Mote 2016 is warranted. e annual frequency of extreme high pressure 'blocking event' days that deliver warm air onto western Greenland peaked in 2010 and 2012 (McLeod and Mote, 2016). Greenland mass loss accelerated between 2003 and 2012 primarily due to increasing surface meltwater runoff (-6.3±1.1 Gt/y2) driven by persistent southerly flow across the western ice sheet (e.g. Rajewicz and Marshall, 2014; McLeod and Mote, 2016). McLeod, J.T. and T.L. Mote, 2016. Linking interannual variability in extreme Greenland blocking episodes to the recent increase in summer melting across the Greenland ice sheet. International Journal of Climatology, 36:1484-1499. Rajewicz, J. and S.J. Marshall, 2014. Variability and trends in anticyclonic circulation over the Greenland ice sheet, 1948–2013. Geophysical Research Letters, 41:2842-2850.

16 20 "A notable exception to the widespread albedo decrease was" io "A notable exception was"

Figures Fig 2, 4,6 Increase text size. In Fig 4 a tiny a, b, c... text is problematic. Figs 3-5 would be an improvement to zoom in to the island of Greenland in each map Fig 4 inset trend map too small? The analysis is v interesting and deserves highlight. May be too many maps compressing the results too much. Remove the grey area. Fig 5 units per day? Small number, multiply to get per month? Fig 7 sorry but I think this analysis does not add sufficiently to the study.

---

## Author Comment (AC1) · 20 Jun 2019

We thank the reviewer for the feedback given. Please see below for a point-to-point response.

albedo can be only used, scientifically speaking, as a proxy of melt extent and melt amount.  Not as proxy of ice discharge (as shown by the authors) and not as proxy of summer SMB (also in part driven by summer snowfall anomalies, as also mentioned by the authors).  Due to the delay between production of meltwater (highly correlated with albedo I agree) and runoff (depending of the snowpack meltwater retention), runoff not be correlated with albedo at the monthly time scale. Using MAR outputs, at the summer time scale, it is true that MJJA SMB is correlated at -0.96 (-0.88 over 1980-1999) with runoff and 0.66 with snowfall over 1980-2015. But in May for example, SMB is correlated at 0.98 with snowfall and -0.13 with runoff, showing well that albedo cannot be used as proxy of monthly SMB, althrough MAR is not "the true".

The reviewer is correct in that snowfall impacts MAR SMB variably, and substantially during the early summer (May). However, MAR model results equally suggest that the impact of May SMB on the overall summer-summed (MJJA) SMB is small, as the mean ice sheet SMB in May is on the order of ~5 Gt, in comparison to the -50…-200 Gt seen in the MJJA sums. Thus, we maintain that while the use of the proxy method to attempt to estimate any single month's SMB is indeed an uncertain venture and cannot be recommended, the proxy method can be useful in its original intended use: to test, on observational basis, whether or not runoff is indeed the dominant driver of the whole summer's summed SMB variability.

In consideration of the reviewer's feedback, we propose to revise the manuscript text in sections 1, 2.7.4., and 3.3 to further clarify and emphasize the intended use of the proxy determination as a test case for runoff dominance at the whole summer scale, to caution the reader against using the proxy to attempt inversion of ice sheet SMB for any single summer month, and to relegate the regression equations to supplementary material to further discourage their careless use.

The albedo data was not used as a proxy for ice discharge in any part of the manuscript. Temporal lag correlation analysis was performed for independent (seasonal) estimates of ice discharge and albedo, as a proxy for meltwater production. However, with regard to the comments here and from the other reviewer, we have removed the time-lag analysis with its associated result figure from the manuscript.

- the role of the albedo decrease and the bare ice expansion to the recent melt increase has already been shown a lot of time in previous publications (e.g. Box et al., 2013).

The reviewer appears to refer to this publication: Greenland ice sheet albedo feedback: thermodynamics and atmospheric drivers, Box et al., 2012. These results, and nearly all others present in literature, are obtained from various versions of the MODIS data record and thus covering 2000->. Recent recalibrations of the MODIS instruments have resulted in a significant update of the referred-to findings (Casey et al., 2017), which the CLARA record is now quite consistent with. Also, as mentioned, this is the first study to leverage the full 1982-2015 coverage in CLARA-A2 albedo, based on intercalibrated AVHRR observations whose stability we demonstrate in the manuscript.

The results also clearly indicate that the significance of albedo trends along the GrIS margins is affected by the length of the observational record being investigated; using CLARA allows us to see the relatively stable 1980s-mid-1990s period, which place the subsequent changes during the MODIS era in larger

temporal context. The manuscript is not a simple replication of past investigations, but rather seeks to extend the viewpoint and contribute to the ongoing discussion.

- the correlation between GBI and melt (approximated with albedo here) is also some-thing which is known from a long time (see the Hanna et al.,....).

The connection between albedo changes and GBI was presented as part of the discussion and not results precisely because it is intended as supporting evidence for what is revealed in the full CLARA period, also for the pre-MODIS era. The discussion around this point has been revised and expanded to provide further viewpoints into the consistency of the albedo decrease with changes in atmospheric circulation.

- the discussion about the cloud cover seem to be out of the purpose of this paper...and using MAR for this is certainly not a robust basis of validation.

MAR cloud cover is not intended as a validation basis for the satellite record, the comparison is presented to highlight the differences between observations and model, and in the context of commentary regarding other recent progress in the field. Cloud cover and cloud optical properties are highly relevant to surface melt and thus albedo changes, the authors do not quite see why their inclusion into the discussion section would be out of purpose.

To conclude, the authors try to correlate their "new" data with several previous stud-ies/estimates but there is no new interesting scientific message in this paper deserving to accept this paper in TC. However, using this new satellite product deserves to be published,  but I recommend to the authors to limit their correlations/comparisons to melt extent (from satellite) or modelled melt amount (as model validation data set).  A comparison with the albedo/bare ice extent MODIS based product (notably used asbare ice albedo in RACMO/HIRHAM) will also be more interesting.

The authors disagree with the reviewer on the lack of novelty value in the manuscript. We show that:

1) albedo changes are significantly negative along many parts of the ice sheet margins, are consistent with MODIS for the overlap, display substantial regional variability in the pre-MODIS era, and are consistent with a variety of atmospheric circulation, air temperature, and cloudiness drivers,
2) bare ice extent has most likely expanded and reaches further up the ice sheet,
3)  snowfall and atmospheric regime anomalies are potentially strong inhibitors of enhanced surface melt,
4) the year-to-year albedo changes and discharge changes are not strongly connected, supporting recent modeling findings.

Finally, the runoff-dominance test for summer GrIS SMB is the final new piece of information offered.

The reviewer's suggestion to focus on melt extent/amount determination is naturally worthwhile as well, but publications in said fields do already exist for the more commonly used microwave observations (Mernild et al, 2011; Välisuo et al, 2018). Extension to the optical domain is certainly possible, but requiring and deserving a separate study in the authors' view.

---

## Author Comment (AC2) · 20 Jun 2019

on comments: a.) io = "instead of" b.) Comments refer to page then line numberusing X Y

General Remark The study has a lot of value by presenting an expanded (in timeand process) treatment of AVHRR albedo over Greenland. The article makes several major analyses, the fourth of which I am not sure should be kept because of its very wide scope, complexity and limited finding.

Thank you for a carefully considered and thorough review. Please see below for our point by point responses. Some typos and unclear expressions were also corrected at the authors' own initiative. All page numbers refer to revised manuscript.

Major critique
A.) The fourth major analysis should be more clearly explained or removed*, that with lag analysis, basin scale examination of hypothetical meltwater lubrication of ice dynamics. The study already has a lot of substance. Adding the lagged result only to confirm earlier studies is a bit much.
12 9-16 reinforces that the study is taking the empirical albedo relation too far.
1 24-27 recommend removing this part of the study as it does not directly examine melt-induced flow acceleration while much has been evaluated more directly on this topic. See SWIPA 2017chapter 6 https://www.amap.no/documents/doc/snow-water-ice-and-permafrost-in-the-arctic-swipa-2017/1610

In line with the feedback here and from the other reviewer, this analysis has been removed from the manuscript.

7 6-11 difficult to follow

Paragraph removed with the albedo-discharge time lag analysis.

B.) Discussion of this study vs Stroeve (2001) Stroeve, J.: Assessment of Greenlandalbedo variability from the advanced very high resolution radiometer Polar Pathfinderdata set, J. Geophys. Res.-Atmos., 106, 33989–34006, doi:10.1029/2001jd900072,2001. How does this study square with Stroeve (2001, see Fig 4 etc) who found de-creasing trends 1981-1998?
8 29 "majority of the albedo decrease signal originates after 2000" but Stroeve 2001 found a decrease before 2000

Stroeve (2001) pointed out that the negative albedo trends detected (at a set of grid points) were not statistically significant and they were largely driven by the anomalously low albedos detected during the summer of 1998. We agree that 1998 was a low-albedo year (see below for CLARA-A2 July monthly mean albedos for 1997-1999), but 1999 was not, therefore the finding by Stroeve might have been different if the following year had been included in the data. Also, our trends are based on the Theil-Sen trend estimator, which is by design robust against outlier influence in the data.

[Figure]

Furthermore, as Polar Pathfinder provides the blue-sky albedo, its seasonal/annual variability is also driven by variability in cloudiness and the cloud radiative properties. Also, the intercalibrations of (earlier) Pathfinder and CLARA-A2 are based on different methods, with the CLARA-A2 method (based on Heidinger et al., 2010) arguably more sophisticated as it leverages the high MODIS calibration as well as stable natural targets. Finally, the Polar Pathfinder dataset only contained data from the afternoon AVHRR satellites, meaning that for the pre-MODIS era, CLARA-A2 has additional observations available from NOAA-12 and NOAA-15 relative to Pathfinder (and additionally NOAA-17, NOAA-19 and METOP-A & B for the MODIS era).

The issue of correctly detecting clouds over bright snow/ice is a consideration for any AVHRR-based study; while some concerns remain on cloud detection accuracy over the high-elevation regions of Greenland (Karlsson et al, 2017), neither the in situ evaluations nor the stability evaluation undertaken here suggest that the large-scale CLARA-A2 ice sheet albedo estimates are significantly influenced by missed clouds. This is likely linked to the leveraging of all AVHRR satellites and the coarsened end product resolution, where typically hundreds or even thousands of reported clear-sky AVHRR GAC-resolution samples are aggregated in a 25 km resolution grid cell to form the grid cell monthly mean albedo. While missed clouds will certainly appear in the data, their impact at the end product scale is ameliorated by the majority of correct clear-sky samples. The 5-day means are more vulnerable to this effect, though, which is partially why statistical Gaussian Process smoothing was applied to the 5-day data in the manuscript.

The spatiotemporal consistency of albedo trends between CLARA and MOD10A1 also reinforces the idea that cloud masking issues are not a dominant driver of the observed trends.

9 19-26 bringing in Stroeve2001 agreement/disagreement seems important here Stroeve found NAO resonance,as one would expect. What about this study?

Please see the remarks above. To make these points to the reader as well, we will include a new paragraph here (pg 10, 7-16) summarizing these differences/likely causes relative to Stroeve (2001). Also, note that some remarks on the cloud masking are also included in the discussion section for clarity.

C.) conclusions . . . 16 12 "The albedo decrease of the northeastern and eastern mar-gins was initiated during the 1982-1999 period". . . you offer a mechanism for the west but what about the east...any idea the cause? It should be either/and atmospheric circulation or sea ice -related.

This is a very good question, to which we presently have no clear answer. The topography of the SE/E coast is quite complex, and while CLARA-A2 SAL does contain a correction algorithm for both geolocation and radiative impacts of mountainous topography (areas with mean slope>5 deg), we remain bound by the overall geolocation accuracy of AVHRR. This implies that we cannot discount sampling errors as a source of influence in complex terrain.

Yet there is some similarity in the negative albedo trends around Blosseville coast in MOD10A1 and CLARA (see next point), some of which are also reproduced by the earlier studies noted here based on various versions of the Pathfinder dataset – although the comparability is limited, as noted in the previous point.

On the other hand, Häkkinen and Rhines (2009) showed that the warm subtropical (surface) waters have begun to penetrate the seas around SE Greenland with increasing intensity, and Straneo et al. (2010) found them present in the Helheim glacier fjord. We could therefore postulate that when the increasing heat energy thus advected on the SE coast is released into the atmosphere, it provides additional energy for the surface melt of snow. This would be consistent with the localized but substantially negative albedo trends seen around Helheim and Kangerdlussuaq glaciers in both MOD10A1 and CLARA-A2. Note that the increasing precip only affects the coast south of Helheim glacier according to MERRA-2.

The case of the NE margins appears different in that oceanic forcing is less likely a cause; we noted that some modeling studies found increasing runoff and thus surface melt, and if downslope winds were increasing along with positive air temperatures, the turbulent flux exchange could also accelerate melt. In this perspective, MERRA-2 does show a statistically significant if unremarkable positive trend in SAT over the NE region (here shown for 78-79.5N, -29 to -32 E, July & August). However, the veracity of the wind fields is untested and thus the quantification of turbulent flux contribution is an uncertain process.

[Figure]

We propose to add new content in the results section (3.1, pg.9 ) and the discussion section (pg 16) summarizing the arguments here.

D.) A direct comparison between albedo data sets: CLARA-A2-SAL and MODIS MOD10A1 seems warranted. How well do they agree in the overlapping period?

We performed a comparison between the datasets, showing the results here for the reviewer's interest, and also propose to add them as supplementary material in the manuscript (Supplementary Figure 4), with commentary in the discussion section (pg 16, 9-17). This is motivated by the fact that a rigorous intercomparison should account for factors which require substantial additional work to quantify, e.g. differences in diurnal sampling, and analysis for the impacts of various downscaling methods to reduce MOD10A1 into the coarse CLARA grid.

We calculated the decadal Theil-Sen albedo trend estimators for the 2000-2015 May-August months for MOD10A1 in its native 5km resolution on the polar stereographic grid (Box et al., 2017, denoised, gap-filled). For the calculation of per-month mean differences during the 2000-2015 overlap, we resampled MOD10A1 to the CLARA grid with a radial weight algorithm with a 25 km radius. While this choice should be broadly acceptable, we note that a more careful intercomparison is deserving of a separate manuscript.

[Figure]

The spatiotemporal distribution of the decadal trends is highly similar. Most of the small-scale albedo decrease features, such as decreases around Helheim and Kangerdlussuaq glaciers, are consistent if limited by the coarse CLARA resolution. The trends in MOD10A1

have larger maxima than CLARA-A2 most likely because of the considerably finer spatial resolution (5 km vs. 25 km) – in CLARA-A2, the narrow regions of sharpest albedo decreases at the (west) margins are smoothed by the spatial aggregation.

The by-month mean difference maps (top row) only show differences above an estimated joint CLARA/MOD10A1 uncertainty envelope of 0.03. As expected considering the results by Alexander et al. (2014), the difference is large in the North during May and August. This difference is stable, though, and does not appear to impact the decadal trends, which agree even for the large-difference regions.

General comments

7 16 agree with "empirically suitable threshold albedo of 0.58" . . . compositing with many PROMICE years yields 0.56 (unpublished)

Thank you, this information is good to know also for future reference.

Recommend to not use abbreviation "GrIS". Instead, use "Greenland ice sheet" until it (very quickly) becomes obvious the study is on Greenland, afterward, use "ice sheet". Should title have"mass balance" io "surface mass balance and ice discharge"

GrIS -> Greenland Ice Sheet revised as suggested throughout the manuscript. However, the title is in our view accurate; the proxy investigation is limited to surface mass balance only, and the manuscript contains a comparative investigation of annual ice discharge and albedo anomalies, even though the time-lag analysis has been removed according to the reviewer's critique.

1 9 "driven in part by " io "primarily driven by "

Revised as suggested.

1 13 "We then subtract ice discharge from the mass change estimates from the GRACE satellite mission to estimate surface mass balance" io "We then correct the mass balance estimates observed by the GRACE satellite mission with state-of-the-art ice discharge"

Revised as suggested.

1 23 "rapid surface mass" io "rapid mass"

Revised as suggested.

2 31 "examining the role of albedo" io "both highlighting and confirming the dominant role of surface melt" would seem to improve the statement by making it not a conclusion placed in the intro of the paper and otherwise clarifying that albedo is the predictor variable here.

Revised as suggested, proposing to amend "albedo" to "albedo-inferred", as albedo is a proxy for surface runoff here.

4 1 "uppermost areas" io "innermost parts"

Revised as suggested.

4 1 - 4 8 . . . Fig3A in Box, J.E., D. van As, K. Steffen, 2017. Greenland, Canadian and Icelandicland ice albedo grids (2000-2016), Geological Survey of Denmark and Greenland Bul-letin, 38, 53-56 available from http://www.geus.dk/DK/publications/geol-survey-dk-gl-bull/38/Documents/nr38_p53-56.pdf supports the idea that 2012 was not anomalously low AND that variability is small (in the Summit GC-Net example; max-min = 0.03) in the dry snow area'

Thank you for the additional reference, added to the text here.

4 12 "~0.02 increase of the GrIS albedo" io "~0.02 overestimation of the GrIS albedo". . . it was a real climate event so the measurement is not an over-estimation

While the reasoning by Stroeve (2001) that the Pinatubo eruption caused cooling which inhibited e.g. Greenland melt for 92-93 is principally valid, the relatively large albedo increase on the top of the accumulation zone in CLARA-A2 for these years (Fig 1) is difficult to fully explain in terms of less surface melt or snow metamorphism– as we normally expect negligible surface melt or metamorphism there anyway. The Pathfinder as well as CLARA records are based on climatological mean aerosol loading over the Arctic – for want of a reliable and spatiotemporally complete aerosol record reaching the 80s – so that some part of the albedo increase could also be explained by unaccounted-for change in atmospheric composition. As the albedo estimates for these years are thus more uncertain than the rest of the CLARA record, we prefer keeping the analysis and text here as is, with some additional explanation for the logic w.r.t. the discussion here (pg 4, 11-21).

8 25 I expect some readers/reviewers will dislike excluding 92 and 93. Yet,I think it's not too questionable as long as you're clear. Here, better I think would be"externally forced" io "less reliable"

Please see the point above. We propose amending the mentioned text to "likely both externally forced and less reliable" to account for both possible explanations.

8 25 "largely remained stable" discuss relative toStroeve 2001

Revised w.r.t. the discussion around major comment B.

8 31 "may be"? Seems more testing needed to address this hypothesis.

Revised w.r.t. the discussion around major comment D, relocated to the discussion section.

9 1 "is" io "may be". . . see/cite Box, J.E., D. van As, K. Steffen, 2017. Greenland, Cana-dian and Icelandic land ice albedo grids (2000-2016), Geological Survey of Denmarkand Greenland Bulletin, 38, 53-56

Relocated to discussion section.

9 5 "rarely examined" "the ice sheet's albedo was primarily stable" see Fig 9c and related discussion in the following where from 1988-1999 eastern Greenland has the largest AVHRR albedo decrease. Some discussion of that seems warranted. Box, J.E., D.H. Bromwich, B.A. Veenhuis, L-S Bai, J.C. Stroeve,J.C. Rogers, K. Steffen, T. Haran, S-H Wang, 2006: Greenland ice sheet surface massbalance variability (1988-2004) from calibrated Polar MM5 output, Journal of Climate,Vol. 19(12), 2783–2800.

Thank you, the authors did not recall that the stated study also contained satellite-based data evaluation. We note that the limitations on Pathfinder/CLARA comparisons as discussed in response to major comment B also likely apply here. However, this reference will naturally be added here and the manuscript revised to reflect these past efforts. The sentence "primarily stable" will be revised to enhance that the finding is only based on CLARA data, and that significant negative albedo trends are apparent on the NE and E margins – in itself the E decreases being consistent with the Pathfinder analysis in the given manuscript.

9 28-34 geolocation errors were attributed in the following study for the relatively noisy ice margin trends. See Box, J.E., D.H. Bromwich, B.A.Veenhuis, L-S Bai, J.C. Stroeve, J.C. Rogers, K. Steffen, T. Haran, S-H Wang, 2006:Greenland ice sheet surface mass balance variability (1988-2004) from calibrated PolarMM5 output, Journal of Climate, Vol. 19(12), 2783–2800.

Certainly they may contribute; the referenced discussion will also be noted as a potential cause of the effect seen. The discussion section now contains the reference as a part of a new paragraph summarizing the MOD10A1-CLARA comparison (pg 16).

10 7 "where the trend signal originates" io "where they are expected to be more robust"

Revised as suggested.

10 10 "larger" io "faster"

Section revised for clarity, phrase removed.

10 21 "earlier" io "faster"

Faster is our preferred term; Figure 7 clearly shows increases in albedo decrease rate (per 30 day period).

10 25 the "increases in winter snowfall" finding is very interest-ing AND is related to the GRACE correlation because when there is snowfall, mass is added and albedo increases. So, be sure to make that point. The following may be relevant if you want to discuss more about how increasing snowfall may be from climate change. https://iopscience.iop.org/article/10.1088/1748-9326/10/11/114008/meta Fur-ther Box et al. (2013) find a climate change signal, an increase in snowfall with NH AirT, N Atlantic Air T, etc. Comparison of Greenland accumulation history with northernhemisphere air temperatures suggests a 6.8% (or 51 Gt) per degree C climate sen-sitivity (Box et al.,

2013). See Box, J. E. 2013. Greenland ice sheet mass balancereconstruction. Part II: Surface mass balance (1840-2010), Journal of Climate,Vol.26, No. 18. 6974-6989. doi:10.1175/JCLI-D-12-00518.1

Thank you, the reference has been added. The work by Wong et al. has been referenced in a new paragraph in the discussion section (pg 13, 16-22) on the NW bare ice exposure changes in context of observed precip trends – which appear quite different from MERRA-2 over the region.

15 20-26 Including discus-sion of Rajewicz and Marshall, 2014; McLeod and Mote 2016 is warranted. e annual frequency of extreme high pressure 'blocking event' days that deliver warm air onto western Greenland peaked in 2010 and 2012 (McLeod and Mote, 2016). Greenland mass loss accelerated between 2003 and 2012 primarily due to increasing surface meltwater runoff (-6.3±1.1 Gt/y2) driven by persistent southerly flow across the western ice sheet (e.g. Rajewicz and Marshall, 2014; McLeod and Mote, 2016). McLeod,J.T. and T.L. Mote, 2016. Linking interannual variability in extreme Greenland blocking episodes to the recent increase in summer melting across the Greenland ice sheet.International Journal of Climatology, 36:1484-1499. Rajewicz, J. and S.J. Marshall,2014. Variability and trends in anticyclonic circulation over the Greenland ice sheet,1948–2013. Geophysical Research Letters, 41:2842-2850.

Thank you, the additional references are added and the raised points noted in the revised manuscript (discussion section, pg 15, 27-34)).

16 20 "A notable exception to the widespread albedo decrease was" io "A notableexception was"

Revised as suggested.

Figures Fig 2, 4,6 Increase text size.

Revised as suggested.

In Fig 4 a tiny a, b, c . . . text is problematic.

Text size increased.

Figs3-5 would be an improvement to zoom in to the island of Greenland in each map

Revised the figures to provide a tighter zoom on Greenland itself.

Fig 4 inset trend map too small? The analysis is v interesting and deserves highlight. Maybe too many maps compressing the results too much. Remove the grey area.

Thank you. The Figure was split into independent figures per period, also providing the inset figure as an independent figure.

Fig 5 units per day? Small number, multiply to get per month?

Revised to reflect change per 30-day period.

Fig 7 sorry but I think this analysis does not add sufficiently to the study

In line with the earlier comment and feedback from Reviewer 1, we have omitted this analysis and the corresponding figures from the manuscript.

---

## Author Response (AR2)

**Editor Decision: Publish subject to minor revisions (review by editor)** (30 Aug 2019) by Mark Flanner

Comments to the Author:

Dear Dr. Riihelä and co-authors -

As you will see from the new referee reports, both reviewers find your revised manuscript substantially improved over the earlier version. At this point, only minor revisions are needed prior to publication. In particular, please:

- take Reviewer #2's advice to remove "ice discharge" from the title, section 3.4 and conclusions, as ice discharge is no longer a focus of the paper.

Title revised; regarding section 3.4., we propose to keep the present short description and results, as they are believed to be useful for the discharge study community, even though the result in question is negative in nature.

- consider reviewer #2's proposal to eliminate section 3.4 (along with comments from Reviewer #1), and make your own decision based on what you think is best for the paper.

Please see above for our reasoning to keep section 3.4., and the associated short mention in the conclusions.

- address Reviewer #1's question about Stroeve [2001, JGR].

Addressed through addition in discussion section, please see the point below for details.

- satisfy Reviewer #1's request to include some discussion of Tedesco et al [2016]

Addressed through addition in discussion section, please see the point below for details.

- consider Reviewer #1's advice for improving the paper by "further emphasizing the main points of the article, perhaps at the expense of un-necessary text"

Restructured discussion and rewrote the final paragraph of conclusions to highlight main findings of study.

- consider Reviewer #1's comments pertaining to specific lines in the text

Each comment taken into account; see below for details.

Thanks, and I look forward to seeing your revised manuscript. -
Mark

**Referee #1**

For final publication, the manuscript should be

accepted subject to **minor revisions**

**Suggestions for revision or reasons for rejection (will be published if the paper is accepted for final publication)**

Some of the litigious results from Sect 3.4 have been removed in this revised version as well as some clear warnings have been added in Sect 2.7.4.

I'm happy now with this version that I suggest to accept for publication after some minor corrections:

- As the main interest of this paper is the presentation of a new satellite data set, I suggest to add at the end of the title
"using the new CLARA satellite-based product"
- I suggest also to remove "and ice discharge" in the title as it is no more presented in the manuscript.

Title revised, "and ice discharge" removed and added "using the CLARA-A2 dataset". As CLARA-A2 is already 2 years old with several existing use cases, though, we prefer not to use the word "new".

- I'm not sure about the useful of keeping Sect 3.4 as most of analysis has been removed. Idem for the sentences about this in the conclusion.

We prefer keeping section 3.4 and a note in conclusions; a negative result is a result nonetheless, though not in the paper's main focus, and the bit of knowledge gained in this short analysis is still believed to be of use to the discharge modeling community.

Referee #2 (J. Box)

For final publication, the manuscript should be

reconsidered after **major revisions**

I am **not** willing to review the revised paper.

**Suggestions for revision or reasons for rejection (will be published if the paper is accepted for final publication)**

Overall synopsis

Another revision further emphasizing the main points of the article, perhaps at the expense of unnecessary text, such as the very last paragraph, would increase the readability of the article.

In the first review, I questioned "Stroeve [2001, JGR] found NAO resonance, as one would expect. What about this study?"… I think that useful question remains unanswered. The summer NAO remains a likely strong predictor of albedo variability. Given the differences in albedo data employed here (black sky) vs blue sky in Stroeve (2001) and thus this topic should be more clearly addressed.

We further analyzed the SAL data against the NAO index (Hurrell et al., 2009) and found a generally similar relationship as did Stroeve (2001); a positive correlation (r=0.47) between NAO and ice-sheet averaged surface albedo. Our correlation was weaker than Stroeve's over the overlap period 1982-1998 (r=0.29), reflecting perhaps the impact of variability in cloud cover in the APP-x blue-sky albedo used in the earlier study. The relationship between GBI and CLARA albedo is stronger, in line with the reasoning in Rajewicz and Marshall (2014). The appropriate paragraph in the discussion section has been expanded to reflect this consideration.

Hurrell, J. W., and C. Deser, 2009: North Atlantic climate variability: The role of the North Atlantic Oscillation. J. Mar. Syst., 78, No. 1, 28-41

Sorry I didn't pick this up earlier but, some solid justification for excluding September from the analysis is justified. The month of May is included. Why not September? The melt season extends into September for at least the southern half of the ice sheet.

CLARA coverage over the ice sheet is more uncertain and only partial during September, because large Sun zenith angles (>70 deg.) prevent reliable estimate calculation. We thus focused only on the months with full coverage. Issue noted in section 2.1.

Sorry I didn't pick this up earlier but, some discussion of the following article is warranted because they also use an AVHRR Greenland product.
Tedesco, M., Doherty, S., Fettweis, X., Alexander, P., Jeyaratnam, J., and Stroeve, J.: The darkening of the Greenland ice sheet: trends, drivers, and projections (1981–2100), The Cryosphere, 10, 477-496, https://doi.org/10.5194/tc-10-477-2016, 2016.

A new paragraph has been added to the discussion section analyzing our findings in relation to Tedesco et al. (2016). The principal results are consistent between GLASS and CLARA, although the role of older impurity exposure is likely less dominant than atmospheric circulation and cloudiness changes, as seen by the effects of the 2013-2015 melt seasons.

Page 17 line 3 "outermost" relative to what?

Removed.

Page 17 line 7, Conclusions … reference is made to "northeastern and eastern margins" then sentence two focuses on western ice sheet driver of albedo variability. So, the sentence could be split to be more clear. Then if no proposed candidate drivers for "northeastern and eastern margins" , state so.

Sentence split and clarified.

Page 17 line 10, Conclusions …"the western and northern ice sheet margins intensified after 2000"… 1998 (and 1995) were warm summers too… I guess MERRA-2 data time series would show that? See Fettweis et al 2007 GRL Fig 2.

Added "primarily" to indirectly acknowledge the 95/98 warm summers.

17 14 "where we propose…" is not a conclusion, is a hypothesis or otherwise speculative, suggest remove from Conclusions and place in discussion.

Done.

17 18 remove the un-needed "state of the art" and "latest" list MAR version number.

Done.

17 24 remove "This…" is speculative and not a conclusion.

Done.

17 28 explain in quantitative terms what is meant rather than "duress" which has a definition of "noun…threats, violence, constraints, or other action brought to bear on someone to do something against their will or better judgment.

See below.

17 30 "precipitation intensity and phase are most likely the strongest drivers of change manifested through…" speculative, remove and stick to firm conclusions of the study. … This whole paragraph seems un-necessary… Rather, I suggest focusing on the main points of the study which are extending the albedo record and what is the value of the AVHRR record, what does it tell us with that many more years of data?… how reliable is the AVHRR period before 2000? Is it possible to make conclusions on how reliable is the AVHRR period before 2000 or does the multi-satellite, vicarious calibration prevent that?

Last paragraph rewritten to focus more on the added value of the record and the consistency of results obtained.